# EvolveDirector: Approaching Advanced Text-to-Image Generation with Large Vision-Language Models

**Rui Zhao**[1], **Hangjie Yuan**[2], **Yujie Wei**[2,3], **Shiwei Zhang**[2], **Yuchao Gu**[1], **Lingmin Ran**[1],
**Xiang Wang**[2,4], **Zhangjie Wu**[1], **Junhao Zhang**[1], **Yingya Zhang**[2], **Mike Zheng Shou**[1,*]

[1]Show Lab, National University of Singapore     [2]Alibaba Group     [3] Fudan University

[4] Huazhong University of Science and Technology

## Abstract

Recent advancements in generation models have showcased remarkable capabilities in generating fantastic content. However, most of them are trained on proprietary high-quality data, and some models withhold their parameters and only provide accessible application programming interfaces (APIs), limiting their benefits for downstream tasks. To explore the feasibility of training a text-to-image generation model comparable to advanced models using publicly available resources, we introduce EvolveDirector. This framework interacts with advanced models through their public APIs to obtain text-image data pairs to train a base model. Our experiments with extensive data indicate that the model trained on generated data of the advanced model can approximate its generation capability. However, it requires large-scale samples of 10 million or more. This incurs significant expenses in time, computational resources, and especially the costs associated with calling fee-based APIs. To address this problem, we leverage pre-trained large vision-language models (VLMs) to guide the evolution of the base model. VLM continuously evaluates the base model during training and dynamically updates and refines the training dataset by the discrimination, expansion, deletion, and mutation operations. Experimental results show that this paradigm significantly reduces the required data volume. Furthermore, when approaching multiple advanced models, EvolveDirector can select the best samples generated by them to learn powerful and balanced abilities. The final trained model Edgen is demonstrated to outperform these advanced models. The code and model weights are available at https://github.com/showlab/EvolveDirector.

## 1 Introduction

In the field of AI-generated content, an increasing number of advanced models have showcased their ability to generate realistic and imaginative images, such as Imagen [1], DALL·E 3 [2], Stable Diffusion 3 [3], Midjourney [4]. While these models benefit from publicly available datasets such as ImageNet [5], LAION [6], and SAM [7], they rely more heavily on proprietary, high-quality data collections that surpass the quality of publicly accessible datasets. Models such as Midjourney [4] are particularly noted for deriving substantial benefits from their internal datasets. However, given the significant commercial advantages brought by their impressive capabilities, most advanced models keep their parameters private, hindering reproducibility and democratization. In this paper, we aim to *explore training an open-source text-to-image model with public resources to achieve comparable capabilities with the existing advanced models.*

---

[*]Corresponding Author.

38th Conference on Neural Information Processing Systems (NeurIPS 2024).

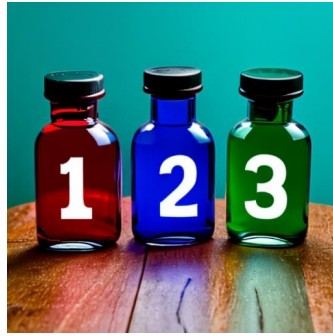

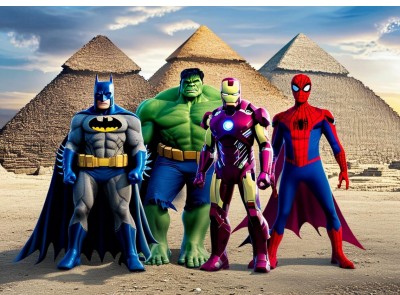

Four superheroes stand in front of the pyramids, from left to right, Batman, Hulk, Iron Man, and Spider-Man.

Three transparent glass bottles on a wooden table. The one on the left has red liquid and the number 1. The one in the middle has blue liquid and the number 2. The one on the right has green liquid and the number 3.

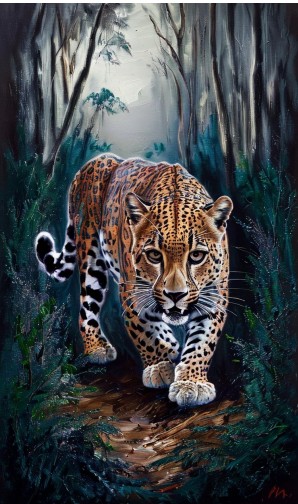

Pallet knife technique, large intentional brushstrokes, masterpiece painting of a leopard silently prowling through the underbrush, dark fantasy.

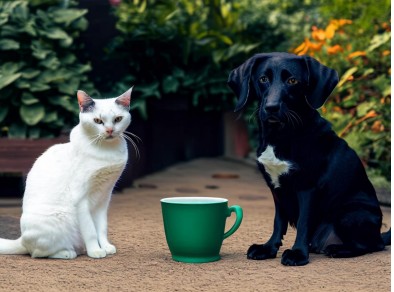

In a garden, there sits a white cat on the left and a black dog on the right, with a green water cup placed between the two of them.

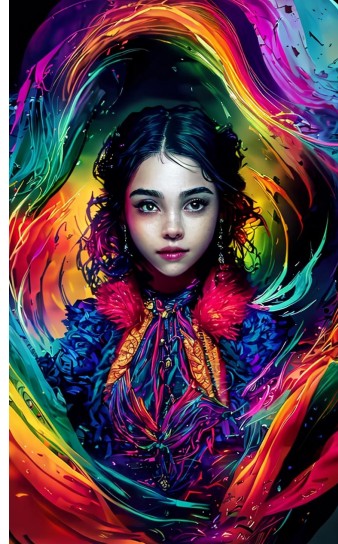

A captivating book cover features a young woman with an enchanting gaze, dressed in vibrant attire, surrounded by swirling colors, inviting readers into a world of chaos and beauty.

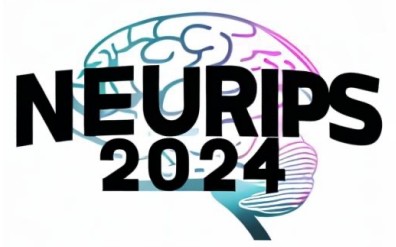

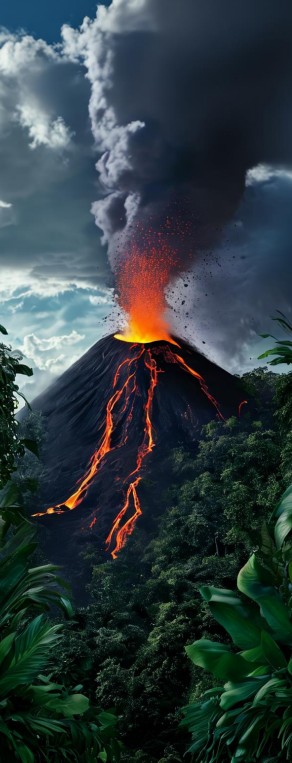

A sleek and modern logo design for NeurIPS 2024. The text "NeurIPS 2024" is written in bold, black capital letters. In the background, there's a minimalist, abstract representation of a brain, with a few key parts highlighted in neon colors. The overall design is clean, visually appealing, and instantly recognizable.

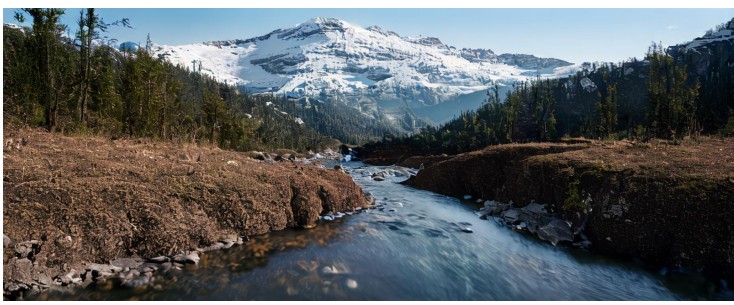

At the foot of the snow-capped mountain, a clear stream winds its way, murmuring and flowing gently, resembling a silver ribbon that connects the forests and canyons.

A volcanic eruption in a dense jungle creates a striking scene as lava flows down the mountain, contrasting with the lush green foliage. Dark clouds fill the sky, casting dramatic shadows over the landscape.

Figure 1: Images generated by our model Edgen (EvolveDirector-Gen). Edgen can generate high-quality images with multiple ratios and resolutions. Notably, it excels in generating text and avoiding attribute confusion when generating multiple objects, which are significant characteristics of the most advanced text-to-image models available today. The input text prompts are annotated under the corresponding images.

Despite the fact that internal data and model parameters of many advanced models remain inaccessible, they provide publicly accessible application programming interfaces (APIs) that enable users to access their generated distribution. This leads to the construction of synthetic benchmarks, *e.g.*, JourneyDB [8] collects 4.7 million images generated by Midjourney [4]. This benchmark is further utilized for enhancing the training of new generative models [9]. However, this paradigm is not data efficient, posing challenges in terms of substantial computation and expenses. Instead of statically constructing multiple expensive large-scale datasets for each advanced model, we take a step forward in this paper by delving into recovering their generative capabilities in a unified framework with limited samples. We propose EvolveDirector to address this challenging task by shedding light on two research questions: (1) *How many synthetic text-image pairs are sufficient to approximate the generative capability of an advanced model?* (2) *Taking it a step further, is it possible for the base model to obtain generative capabilities beyond the advanced models?*

To explore the first question, we start with a demonstration experiment by training a relatively poor model, a DiT model [10] pre-trained on public dataset ImageNet [5] and SAM [7], to approach the advanced model PixArt-$\alpha$ [9] using increasing data scales. The training data is curated by collecting diverse text prompts and utilizing them to generate images from PixArt-$\alpha$. The experiments indicate that when we scale the training data (*i.e.*, generated image-text pairs) to 11 million, we can obtain a base model achieving similar capabilities to the target model without access to its internal data. However, the magnitude of 11 million generated data is comparable to the 14 million data used for pre-training the target model. Training a base model in this way incurs significant expenses, not only in terms of time and computational resources but also the costs associated with using fee-based APIs of some advanced models.

For more efficient training, it is crucial to minimize data redundancy and maximize data quality, as the marginal benefit of training on inferior data is limited. The corpus of the 11 million text prompts, generated from the SAM dataset [9], and the images generated by the advanced model exhibit redundancy in several aspects: (1) *lacking imaginative text prompts* due to the photographic nature of SAM images; (2) *high similarities among text prompts*; and (3) *imbalanced data quality*. The generated images using the target advanced model may exhibit low quality due to inferior alignment on some text prompts. To address these problems, we introduce large vision-language models (VLMs) to improve the diversity and quality of training data for efficient training. Our approach involves a continuous evaluation with VLMs to dynamically refine the training dataset by the discrimination, expansion, deletion, and mutation operations. This dynamic curation strategy ensures that only valuable data is retained, significantly reducing the volume of data for training. Experimental results demonstrate that a mere 100k training samples are sufficient for the base model to gain similar performance to that of the target model, which is substantially fewer than the 11 million samples required by the baseline method.

To explore the second question, we applied EvolveDirector to train the base model to approach multiple recent most advanced models in a unified framework, including DeepFloyd IF [11], Playground 2.5 [12], Stable Diffusion 3 [3], and Ideogram [13]. For each text prompt, we invoke advanced models to generate their images, and the VLM selects the best match to train our base model. The final trained model is named Edgen (EvolveDirector-Gen). The experimental results demonstrate that Edgen outperforms the advanced models mentioned above. Although our initial goal was to approximate these advanced models, we ultimately benefited from the VLM in choosing better training samples from their generated data, thereby achieving capabilities superior to any individual model.

The code of EvolveDirector and the model weights of Edgen are released to benefit the downstream tasks. Our contributions are summarized as: (1) Through experiments on massive data, we conclude that the generation abilities of a text-to-image model can be approximated through training on its generated data. (2) We propose the EvolveDirector, a framework that harnesses VLM to direct the training of the base model to learn generation ability from advanced models efficiently. (3) The trained text-to-image model Edgen outperforms the most advanced models.

## 2   Related Works

**Text-to-Image Generation.** To advance the overall quality of text-to-image generation, research efforts have been invested in exploring architectural improvement [10, 14, 15, 16] and generation paradigm advancement [17, 18, 3], *etc*. Diffusion models stand out as the de facto text-to-image

generation paradigm [19, 20, 1, 3, 21, 22, 23, 24, 25], noted for its scalability and stability [26]. They benefit numerous downstream tasks, spanning image editing [27, 28], video generation [29, 30, 31, 32], 3D content generation [33, 34], *etc*. Through the use of highly descriptive and aligned image-text pairs at a substantial scale, text-to-image models that excel in resolutions, safety control, and the capability to render accurate scenes are obtained, *e.g.*, Imagen [1], Midjourney [4], DALL·E 3 [2], Stable Diffusion 3 [3], and Ideogram [13]. However, the exceptional capabilities of most advanced models have led providers to restrict access, typically offering only APIs, which limits their widespread and equitable use. In this paper, we aim to fill this gap by leveraging advanced VLMs to direct base model to replicate the functionality of advanced models. In contrast, some works propose to motivate models to learn from their self-generated images [35, 36].

**Evaluating T2I Generation with VLMs.** Some automatic evaluation methods [37, 38, 39] are propoesd to combine the LLMs and VQA models to evaluate the generated contents. Thanks to the substantial advancement of large language models [40, 41, 42, 43, 44, 45], the capabilities of VLMs are largely boosted [46, 47, 48, 49, 50, 51]. Utilizing these enhanced capabilities, methods building on VLMs are designed to facilitate the evaluation of text-to-image generation [52, 53, 54, 37, 55]. Notably, the recent research effort, Gecko [54], demonstrates the practicality of leveraging pre-trained LLMs [44] and VLMs [45] for systematic evaluation of text-to-image generative performance, spanning aspects such as text rendering, relational generation, attribute generation, *etc*. However, previous research requires fine-grained evaluation across various aspects, which remains challenging. In EvolveDirector, we simplify this process by requiring only pairwise comparisons, which enables a stable and reliable performance, facilitating the approaching of advanced text-to-image models.

**Knowledge Distillation.** KD [56] aims to transfer knowledge from well-trained teacher models to a simpler student model. The primary focus of most research in KD lies in investigating distillation losses with the output predictions of teacher model [57, 58], intermediate feature maps [59, 60], or feature correlations [61, 62] to distill knowledge. Recently, distillation methods based on diffusion models have garnered attention, primarily by distilling outputs from intermediate steps of the diffusion process to expedite the sampling process [63, 64, 65, 66]. Despite sharing the same goal of approaching the performance of advanced models, we would like to highlight our training paradigm is an orthogonal procedure to KD. To approach the advanced models with only APIs available, we avoid the need for distillation losses or acquiring intermediate results and instead choose a data-driven paradigm. Furthermore, our designed paradigm is also orthogonal to dataset distillation [67] as we evaluate and refine data during training rather than relying on a pre-existing large dataset.

**Online Learning.** Online learning has garnered significant interest for its capacity to enable models to adapt to real-time and dynamic data scenarios [68, 69]. This attention extends to a variety of real-world tasks, including semi-supervised learning [70, 71], unsupervised learning [72, 73], and continual learning [74, 75, 76], *etc*. In EvolveDirector, we harness the potential of online learning and powerful VLMs to evolve models towards advanced generation models, offering an efficient and scalable framework capable of dynamically adapting to evolving data.

## 3 Method

In this section, we outline the EvolveDirector framework in Sec. 3.1, describe the detailed operations of the VLM within this framework in Sec. 3.2, and discuss the training strategies in Sec. 3.3.

### 3.1 Overview of EvolveDirector

The proposed framework EvolveDirector, as shown in Fig. 2, comprises three parallel processes: (1) interacting with advanced T2I models to get training images, (2) maintaining the dynamic training set empowered by the Vision-Language Model (VLM), (3) training the base model on the dynamic training set. The dynamic training set is updated by the VLM and advanced T2I models, to ensure the data are high value for training, thereby achieving efficient training and reducing the overall required data volume.

**Interaction with Advanced Models.** While the model configurations and weights of many advanced T2I models are not publicly accessible, they often provide APIs for interactions. In EvolveDirector, we interact with these APIs by submitting text prompts and receiving the corresponding generated images. The one that aligns with the given text prompt better will be selected by the VLM and

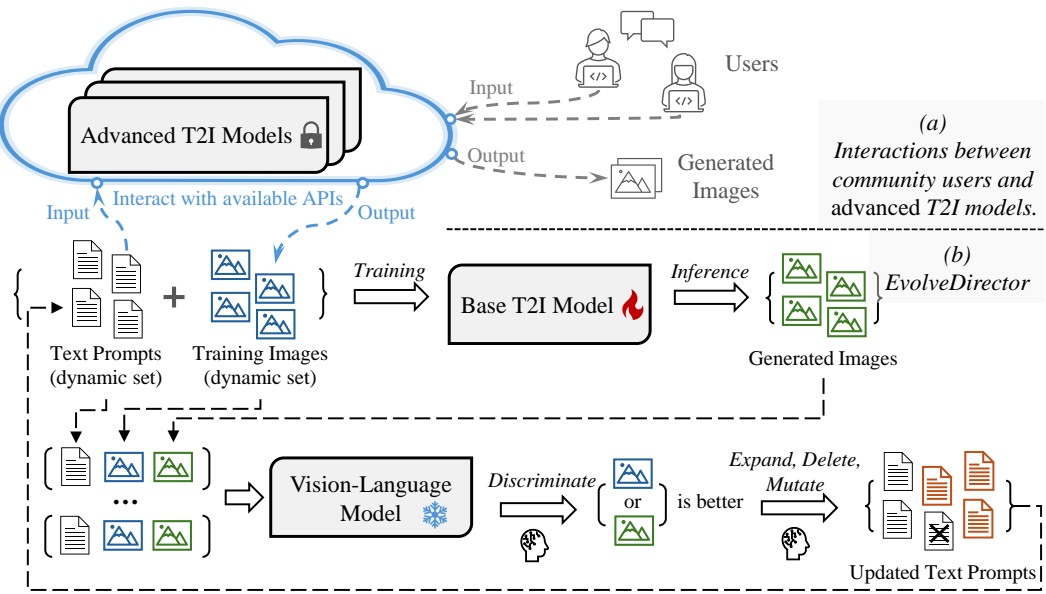

Figure 2: The overview of the proposed framework EvolveDirector. (a) Advanced T2I models provide accessible APIs, allowing users to input text prompts and get the generated images. (b) The base model is trained on the dynamic dataset, consisting of text prompts and corresponding images generated by advanced models via API calls. The VLM continuously evaluates the base model and, according to its performance, dynamically updates and refines the dataset through discrimination, expansion, deletion, and mutation operations based on its evaluations.

included in the training set. The selection criteria and details of these advanced models are elaborated in Sec. 4.4.

**Dynamic Training Set.** The training set is dynamically updated during the training of the base model. It continuously incorporates high-value samples, *i.e.* text prompts on which the base model underperforms compared to advanced models. Simultaneously, it dynamically excludes low-value samples, *i.e.* those on which the base model performs comparably to the advanced models. The VLM evaluates the value of samples, with a detailed procedure outlined in Sec 3.2. The advanced T2I models are continuously called to generate new images for the new text prompts and update them into the dynamic training set.

**Training of the Base Model.** Based on the dynamic training set, we train a Diffusion Transformer (DiT) [10] as our base text-to-image model. Specifically, the model is built upon the improved architecture proposed by PixArt-$\alpha$ [9]. Besides, we incorporate layer normalization after the Q (query) and K (key) projections in the multi-head cross attention blocks [77] to further stabilize the training, $\text{Attention}(Q, K, V) = \text{softmax}\left(\frac{f_Q(Q) \cdot f_K(K)^T}{\sqrt{d_k}}\right) V$, where $f_Q(\cdot)$ and $f_K(\cdot)$ are the layer normalizations after the projections.

### 3.2 Vision-Language Model as Director

The VLM acts as a director to guide the construction of more valuable dynamic datasets. To simplify the task difficulty and better leverage the capabilities of VLM, we present it with a choice of two images as a multiple-choice question, as shown in the top left corner of Fig. 3. One image $I_{advanced}$ is generated by the advanced model, while another one $I_{base}$ is generated by the base model, respectively. In practice, the order of the two images is randomized. VLM is called to compare them and decide which one aligns better with the given text prompt $T$. Regarding the choice of VLM, there are two potential scenarios as follows.

(1) $I_{advanced}$ outperforms $I_{base}$. The inferior performance on this text prompt indicates that the base model is still under-trained. Therefore, EvolveDirector utilizes the VLM to generate more $N_S$ variations of this text prompt $T$, as shown in the lower-left corner of Fig. 3. Then the advanced T2I

model generates corresponding images to expand the dynamic training set, as shown in the right side of Fig. 3. The original samples will continuously be involved in the training.

(2) $I_{advanced}$ does not outperform $I_{base}$. If the base model is comparable with the advanced model, indicating sufficient learning, the VLM will remove that prompt $T$ as a low-value sample from the set to economize on training resources.

Besides the expansion and deletion operations, EvolveDirector also performs mutation operations with a certain probability. This operation permits the VLM to generate more diverse text prompts independent of any existing ones, thereby encouraging the model to explore and learn from a broader domain of text prompts.

### 3.3 Training Strategies

**Online Training.** To boost training efficiency, we develop EvolveDirector as an online training framework. Specifically, the base model undergoes uninterrupted training, without pausing for the advanced model or the VLM to execute. Instead, it sends a command to start VLM evaluation every 100 epochs, termed a checking epoch. At the checking epoch, a subset of text prompts is sampled from the dataset with a specific ratio $R_S$. They are fed into a replica of the base model to generate images. Then the VLM evaluation and the subsequent execution of EvolveDirector begin, as illustrated in Fig. 3. Finally, a command is sent to the trainer of the base model to update the dynamic dataset. A detailed analysis of hyperparameters, including select ratio $R_S$ and extension number $N_S$, is provided in the supplementary.

**Stable Training.** In our task setup, the scale of training data is significantly less than the million

Figure 3: An example of the interaction between the EvolveDirector, VLM, and advanced T2I model. For brevity, auxiliary instructions to the VLM are omitted in this figure.

scale. Under this circumstance, the original architecture [9] demonstrates considerable instability during training, and the generation collapse is observed. Thus we follow the work [77] and apply the layer normalization after the query and key projections to improve the training stability. The experimental results detailed in the supplementary demonstrate the effectiveness of this adaptation.

**Multi-Scale Training.** The ability to generate images across various scales and aspect ratios is a significant capability of advanced T2I models. To facilitate more efficient training, we initially train the base model on images with a fixed resolution of 512px. Subsequently, we extend the training to images of higher resolution with multiple aspect ratios, thereby enabling the model to generate multi-scale and multi-ratio images. Following [9], we construct buckets with different aspect ratios and image sizes. In EvolveDirector, for each text prompt, a size bucket is randomly sampled from the buckets and advanced T2I models are called to generate images with this size. To avoid generation collapse, if the selected bucket falls outside the optimal size range of the advanced model, it will be resized to the closest appropriate size.

## 4 Experiments

### 4.1 Training Details

We train the base model on 16 A100 GPUs for 240 GPU days, with a batch size of 128 and 32 for images at 512px and 1024px resolution, respectively. The VLM evaluation process is distributed across 8 A100 GPUs to facilitate its speed. The open-source advanced models are deployed on our

devices with simulated APIs for interaction. For closed-source models, EvolveDirector interacts with them through their public APIs.

## 4.2 Selection of Vision-Language Models

The ability of the VLM to analyze images is crucial to the EvolveDirector framework. There are multiple powerful VLMs available, including CogVLM [82], CogAgent [78], Qwen-VL (Qwen-VL-Chat, Qwen-VL-Plus, and Qwen-VL-Max) [42], InternVL [79], LLaVA and LLaVA-Next [51, 80], and GPT-4V [81]. We evaluate their newest version on 600 pairs of questions to calculate the alignments with human raters. Each question consisted of a text

Table 1: Alignment of VLMs with Human Preferences. The best value is highlighted in blue, and the second-best value is highlighted in green.

| | Discrimination | Expansion Accuracy | Diversity |
|---|---|---|---|
| CogAgent [78] | 0.675 | 3.81 | 3.76 |
| Qwen-VL-Max [42] | 0.825 | 4.72 | 4.56 |
| InternVL [79] | 0.793 | 4.70 | 3.67 |
| LLaVA-Next [80] | 0.840 | 4.85 | 4.69 |
| GPT-4V [81] | 0.847 | 4.82 | 4.72 |

prompt and two images generated based on that prompt, presented to 5 different human raters. The VLMs are tested in two aspects: (1) Discrimination, and (2) Expansion. To be more specific, (1) initially, the VLMs were required to select which image aligned more closely with the given text prompt. The output is scored by 0 for wrong or 1 for correct by human raters. (2) Subsequently, they are instructed to generate more variations of the given text prompt. The score of "Accuracy" evaluates whether the generated text prompts contain any linguistic errors and whether they are in the same syntactic structure as the given text prompt. The score of "Diversity" evaluates the diversity among the generated text prompts. These two scores range from 1 (worst) to 5 (best). The results are shown in Tab. 1. The LLaVA-Next and GPT-4V achieve the top performance. Considering that LLaVA-Next is totally free to use, we select it as the VLM for EvolveDirector.

## 4.3 Ablation Studies

**Models.** For ablation studies, we select the Pixart-$\alpha$ [9] as the unified advanced model to approach. We utilize a DiT model pre-trained on publicly accessible data, *i.e.* the ImageNet and SAM dataset, as the base model.

**Metrics.** We sample $10,000$ text prompts to feed into models trained under different ablation settings to generate images and calculate their FIDs with the images generated by the advanced model. These text prompts are not seen by these

Table 2: Ablation Studies. The best value is highlighted in blue, and the second-best value is highlighted in green.

| Discrimination | Expansion | Mutation | Data Scale | FID ↓ | Human Evaluation ↑ |
|---|---|---|---|---|---|
| ✗ | ✗ | ✗ | 11M | 7.36 | 48.89 |
| ✗ | ✗ | ✗ | 1M | 11.49 | 39.44 |
| ✗ | ✗ | ✗ | 100K | 15.19 | 32.22 |
| ✓ | ✗ | ✗ | 100K | 15.41 | 32.61 |
| ✗ | ✓ | ✗ | 100K | 13.05 | 36.44 |
| ✓ | ✓ | ✗ | 100K | 7.61 | 47.00 |
| ✓ | ✓ | ✓ | 100K | 7.45 | 48.53 |

models in the training stages. To conduct human evaluation, we randomly selected 300 sets of text prompts and their corresponding generated images, which are $2,400$ in total. These images are paired in twos, each consisting of one image from the advanced model and one from an ablation model corresponding to the same text prompt, arranged in random order. Then they are presented to 5 different human raters to choose which image matches the text prompt better. In this manner, each ablation model is paired with the advanced model for comparison, and their selected ratios are recorded to reflect their relative performance compared to the advanced model.

**Results.** The results of FIDs and human evaluation are shown in Tab. 2. (1) *Directly Training on Generated Data.* The first three models were directly trained on images generated by the advanced model, using image quantities of 10 million, 1 million, and 100 thousand respectively. The experimental results show that the model trained on 10 million data reaches a comparable level to the advanced model in terms of human preference ($48.89\%$ V.S. $51.11\%$), and achieves a low FID score (7.36). This indicates that the generative capabilities of the advanced model can be learned through training on its large-scale generated data. However, if the training data is reduced to $10\%$ or even $1\%$ of the original amount, the performance of the trained model will significantly decrease.

(2) *Training with EvolveDirector.* In the last four rows of Tab. 2, models trained with EvolveDirector under different ablation settings are evaluated. The upper bound of data volume is set to 100k for all models. The first model is trained on an initial number of 100K data and dynamically deletes data. The last three data models start training on an initial number of 2K data and dynamically

add new data to learn. The first model applies the *"Discrimination"* function of the VLM model to discriminate the generated samples of the base model. Samples comparable to the output of the advanced model are removed from the training dataset. The results show that dynamically deleting samples does not cause much performance degradation. The second model does not use the VLM to evaluate the base model but instead randomly selected training samples for *"Expansion"*, i.e. generating more variants of given prompts and training samples. The results show that this operation brought a slight performance improvement due to the expansion of text prompt diversity. The third model utilizes VLM to evaluate the base model and performs reasoned expansion and deletion based on the evaluation results. As the results indicate, there is a significant performance increase, which highlights the importance of the evaluation of VLM. Lastly, the complete version of EvolveDirector is tested, which further applies the *Mutation*, i.e. randomly generating entirely new text prompts with a $10\%$ probability. This operation further improves the performance of the model, because it encourages the model to explore more diverse images. With the full version of EvolveDirector, the model trained on a dynamic dataset with a data cap of 100K, achieved performance comparable to the model trained on 10M generated data, indicating that the proposed framework could significantly reduce the amount of training data required to approach the performance of the advanced model.

It is worth noting that there is still a slight gap between the final model and the advanced model, which can be attributed to the law of diminishing returns. This occurs because it becomes increasingly difficult to identify truly high-value samples as the performance approaches that of the advanced model. However, this phenomenon vanishes when EvolveDirector learns from multiple advanced models simultaneously, as experiments in Sec. 4.4 demonstrated. This is because the larger performance gaps between multiple models facilitate the easier identification of high-value samples.

## 4.4 Approaching Advanced Models

We select several latest advanced models to approach their powerful generation ability, including Playground 2.5 [12], Stable Diffusion 3 [3], and Ideogram [13]. Playground 2.5 is famous for its aesthetic generative effects, while Stable Diffusion 3 and Ideogram are known for their strong performance in various aspects including text generation and multi-object generation. Be-

Table 3: Select Ratios of Advanced Models. The highest value is highlighted in blue , and the lowest value is highlighted in gray .

| | Overall | Specific Domain | | |
| --- | --- | --- | --- | --- |
| | | Human | Text | Multi-Object |
| DeepFloyd IF [11] | 18.57% | 9.12% | 21.21% | 12.12% |
| Playground 2.5 [12] | 25.71% | 31.33% | 9.09% | 25.24% |
| Ideogram [13] | 29.52% | 29.18% | 34.24% | 33.36% |
| Stable Diffusion 3 [3] | 26.19% | 30.36% | 35.45% | 29.27% |

sides, we select a relatively old model, DeepFloyd IF (but just released in April 2023) [11], for its amazing text generation ability. The base model is the same as the one in Sec. 4.3, and we continue to train the base model that has already approached the Pixart-$\alpha$ to approach the selected multiple advanced models. During training, EvloveDiretcor will feed each text prompt to all of them to generate corresponding images, and then use VLM to select the best one of them as the image from the advanced model. The selected image then undergoes evaluation as detailed in Sec.3.2.

**Select Ratios of Advanced Models.** We have calculated the proportions of images generated by these models that were selected by the VLM in the training stage, as shown in Table 3. Among the generated images, those generated by Ideogram are selected with the highest ratio. Besides, we evaluate the select ratios of images in specific domains, such as human generation, text generation, and multiple object generation. Examples of these three types are shown in Fig. 5. The results in Table 3 show that different advanced models achieve the highest select ratios in different specifics. For example, for generating text in images, the Stable Diffusion 3 outperforms others, while the Playground 2.5 is much worse than others. This may be caused by that the internal training data of Playground 2.5 does not include sufficient high-quality images with text in them. This also demonstrates the significance of diverse high-quality data.

**Quantitative Comparison.** To evaluate the performance of the trained Edgen, the base model, and the advanced models, we sample text prompts and feed them into each model to generate images. One image generated by Edgen and one image generated by the comparison model are combined together. Finally, same with the scale of comparison in previous works [9], 300 text prompts and 1800 image combinations are shown to human raters to select which one aligns with the given text prompt better. Each combination is evaluated by 5 different human raters. The results are shown in Fig. 4. We first analyze the performance of each model across all tested text prompts, as shown on the left side of Fig. 4. It is noteworthy that during the training, the VLM selects the best ones among

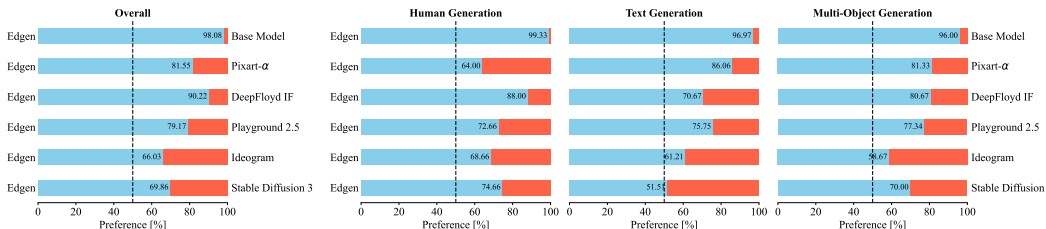

Figure 4: Human evaluation of the images generated by the base model, Edgen trained by the proposed EvolveDirector, and multiple advanced models.

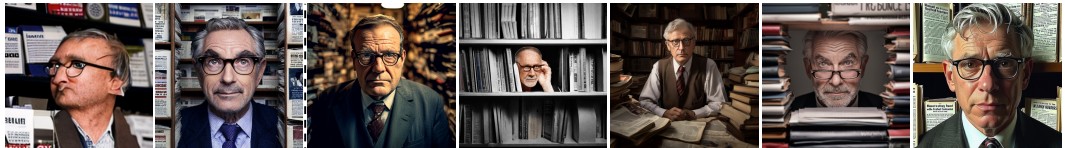

"A seasoned journalist, his charismatic face framed by shelves of archived stories and reports. His eyes, sharp and scrutinous, glance over the rim of his glasses at the historic headlines that surround him."

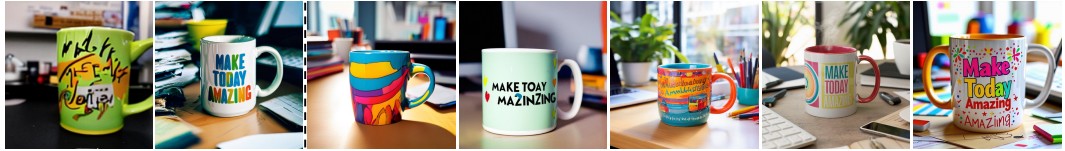

"A cheerful mug on a cluttered office desk, its side emblazoned with 'Make Today Amazing' in vibrant, uplifting colors."

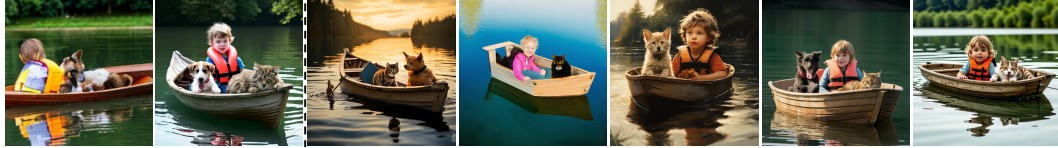

"A small wooden boat drifting on a calm lake, containing a young child wearing a life jacket, a playful puppy with its tongue out, and a calm old cat dozing off beside them."

Figure 5: Images generated by the base model, Edgen trained by our EvolveDirector, and multiple advanced models. The results in three rows showcase the generation of human, text, and multi-object.

the images generated by multiple advanced models to be used as training data. Therefore, although EvolveDirector initially aims to train the base model to approach them, the final trained model Edgen outperforms all of the advanced models. Besides, we evaluate the performance of these models on specific types of text prompts, with the results displayed on the right side of Fig. 4.

**Qualitative Comparison.** In Fig. 5, we showcased three groups of generated images. The three rows of results respectively demonstrate the generation capabilities for humans, text, and multiple objects. For the first group, only the images generated by Edgen, DeepFloyd IF, and Ideogram successfully reflect the "face framed by shelves of ...". As shown in the second row, only Edgen, Ideogram, and Stable Diffusion 3 have the ability to generate correct text in images. For the results shown in the third row, three objects need to be generated, i.e. the child, puppy, and cat. Only Edgen and Ideogram success and other models lost some objects. These results show that Edgen has already learned powerful generation abilities and outperforms some advanced models.

# 5   Conclusion

In this paper, we propose EvolveDirector, a framework that targets approaching the generation capabilities of advanced text-to-image models by only utilizing their publicly accessible APIs. By harnessing the capabilities of large vision-language models for evaluating image-text alignment,

EvolveDirector significantly reduces the volume of training data required, thus saving considerable training costs, especially those associated with API usage. Experimental results demonstrate that the resultant model Edgen, inheriting the generation capabilities from multiple advanced models, achieves superior performance in various aspects. The limitations and future work are discussed in the supplementary.

## Acknowledgements

This project is supported by the Mike Zheng Shou's Start-Up Grant from NUS. It was partially done by Zhao Rui during his internship at Alibaba Group, and Zhao Rui is partially supported by the Alibaba Research Intern Program.

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

## A    Limitation and Future Work

While EvolveDirector achieves significant strides in approximating the generation capabilities of advanced text-to-image models, the resultant model can face challenges related to bias. Our model can inadvertently inherit biases present in the images generated by the advanced models. Additionally, our reliance on VLMs to evaluate generated images could introduce their own biases into the selection process. To mitigate these issues, it is better to integrate debiasing methods and incorporate human feedback in future developments, aiming to enhance the robustness and fairness of EvolveDirector.

## B    Broader Impacts

Our research leverages the foundational capabilities of large vision-language models to reproduce advanced text-to-image models. This initiative represents a pioneering approach to make state-of-the-art generative technologies more accessible and cost-effective. Our methodology substantially reduces the volume of data required for model training, thereby reducing computational demands and minimizing environmental impacts associated with image generation. The resultant model Edgen has the potential to revolutionize the creation of digital media owing to its inheritance of capabilities from multiple advanced models. For instance, its capabilities enable the creation of diverse media content and marketing materials, such as logos, by effectively rendering text into realistic and imaginative visual representations.

However, we also acknowledge potential negative societal impacts. As with many generative technologies, there is an inherent risk of bias in terms of gender and race in the generated content. Furthermore, the improper use of the resultant model, such as inputting harmful or obscene text prompts or creating unauthorized or deceptive images of public figures, could facilitate misinformation or impersonation. To mitigate these risks, it is imperative to implement thorough oversight to secure its ethical use.

## C    Layer Normalization

As discussed in the main paper, due to the scale of training data being significantly less than the previous million scale, the original architecture [9] is not stable during training. As shown in Fig. A (a) and (b), when the training steps increase from 0.5k to 2k, the main object in the generated image begins to be destroyed. To address this problem, we incorporate layer normalization in the multi-head cross-attention. To be more specific, the layer normalization is added after the Q (query) and K (key) projections. This can significantly stabilize the training of the base model, Training model with the layer normalization can avoid generating destroyed images, as shown in Fig. A (c).

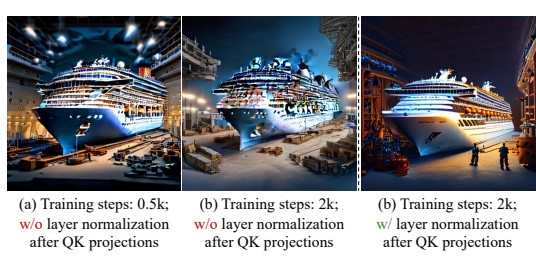

(a) Training steps: 0.5k; w/o layer normalization after QK projections

(b) Training steps: 2k; w/o layer normalization after QK projections

(b) Training steps: 2k; w/ layer normalization after QK projections

Figure A: Illisturation of the impact of incorporating layer normalization after QK projections on generation.

## D    Detailed Instructions to VLM

**Instruction for Discrimination.** We input the combination of two images and instruct the VLM to choose which one aligns with the given text prompt better. The instruction is as follows, "In these two images (A) and (B), which one aligns better with the text description "*text prompt*"? You have two options: <(A) is better>, <(B) is better>. Simply state your choice, no need for a detailed explanation."

**Instruction for Expansion.** The instructions for expansion are as follows, "Replace the nouns in the text description: "*text prompt*" with other kinds of objects, characters, backgrounds, names, colors, or styles to generate *extend num* more diverse text descriptions. Arrange them in the format of a list ["Text description 1", "Text description 2", ...]."

**Instruction for Mutation.** The instructions for expansion are as follows,

"Now, exercise your imagination to generate one new text description for visual content that is completely unrelated to the previous images. It should have a completely different structure from the previous descriptions. *enhanced prompt*. Arrange it in the format of a list just like ["xxxxx"]."

The *enhanced prompt* is randomly sampled from the following ones:

- "It should be rough and short."
- "It should contain less than 30 words and be highly detailed."
- "It should contain over 30 words with different granular levels of detail."
- "It should contain over 50 words with a lot of details."

**Structure the Outputs of VLM.** The diversity of output formats from VLM can pose challenges for automated parsing. We found that by providing specific instructions to the VLM, its output format can be standardized. Specifically, when prompting the VLM to generate more text prompts, we offer instructions such as "*Arrange them in the format of a list ["Text description 1", "Text description 2", ...].*" This approach directs the VLM to generate outputs in a consistent format.

# E  Hyper-parameters Setting

The hyper-parameters of EvolveDirector include the select ratio of images selected from the dynamic training set for evaluation (select ratio $R_S$) and the number of training samples expanded based on a single sample (number of extensions $N_E$). These two parameters mainly affect the growth rate of training samples in the dynamic training set. Too fast a growth rate can lead to the need to generate a large number of images in the early stages of training, thereby quickly increasing training costs, while too small an increase can cause the model to explore the diversity of different samples too slowly. To balance the training costs and the diversity of samples explored by the model, it is necessary to set reasonable values for $R_S$ and $N_E$. Due to the high cost of searching for the most reasonable parameter combination (fully training the model under each combination requires 240 A100 GPU days), we only explored a limited number of parameter combinations and stopped the training as long as we observed the trend in data growth.

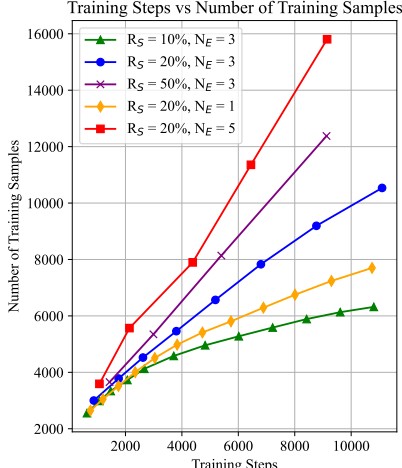

Figure B: The effect of values of hyper-parameters on training data growth rate.

As shown in Fig. B, five kinds of combination of select ratio $R_S$ and number of extensions $N_E$ are evaluated, we can observe that if the $R_S$ is too large, the volume of training data increased rapidly with increasing cost too fast. Despite a larger number of extensions $N_E$ encouraging the model to explore new training samples greedily, it also increases the training cost rapidly. In practice, we found that setting it to 3 is enough to explore new samples. Finally, we select the combination of $R_S = 20\%$ and $N_E = 3$ to achieve a good balance of training cost and exploration of diverse samples. Besides, since the mutation rate $R_M$ does not affect the growth rate of the training set significantly, we simply set it to $10\%$ as an augmentation of the exploration of samples.

# F  Implementation Details

The model is trained by the AdamW optimizer with a learning rate of $2e-5$ and with a gradient clip of 1.0. A constant learning rate schedule with 1000 warm-up steps is used. The gradient checkpointing is used to save the VRAM. For generating 512px images, we train the base model with 100K training steps, and for 1024px image generation, we train the model with 20K training steps.

The default number of training samples at the beginning of training is 20K, which will gradually grow to the upper bound. For the ablation studies in Sec. 4.3, we set the upper bound to 100K to explore the extreme performance of EvolveDirector. The initial text prompts are sampled from those captioned for the SAM dataset [9]. For approaching multiple advanced models in the Sec.4.4, we set

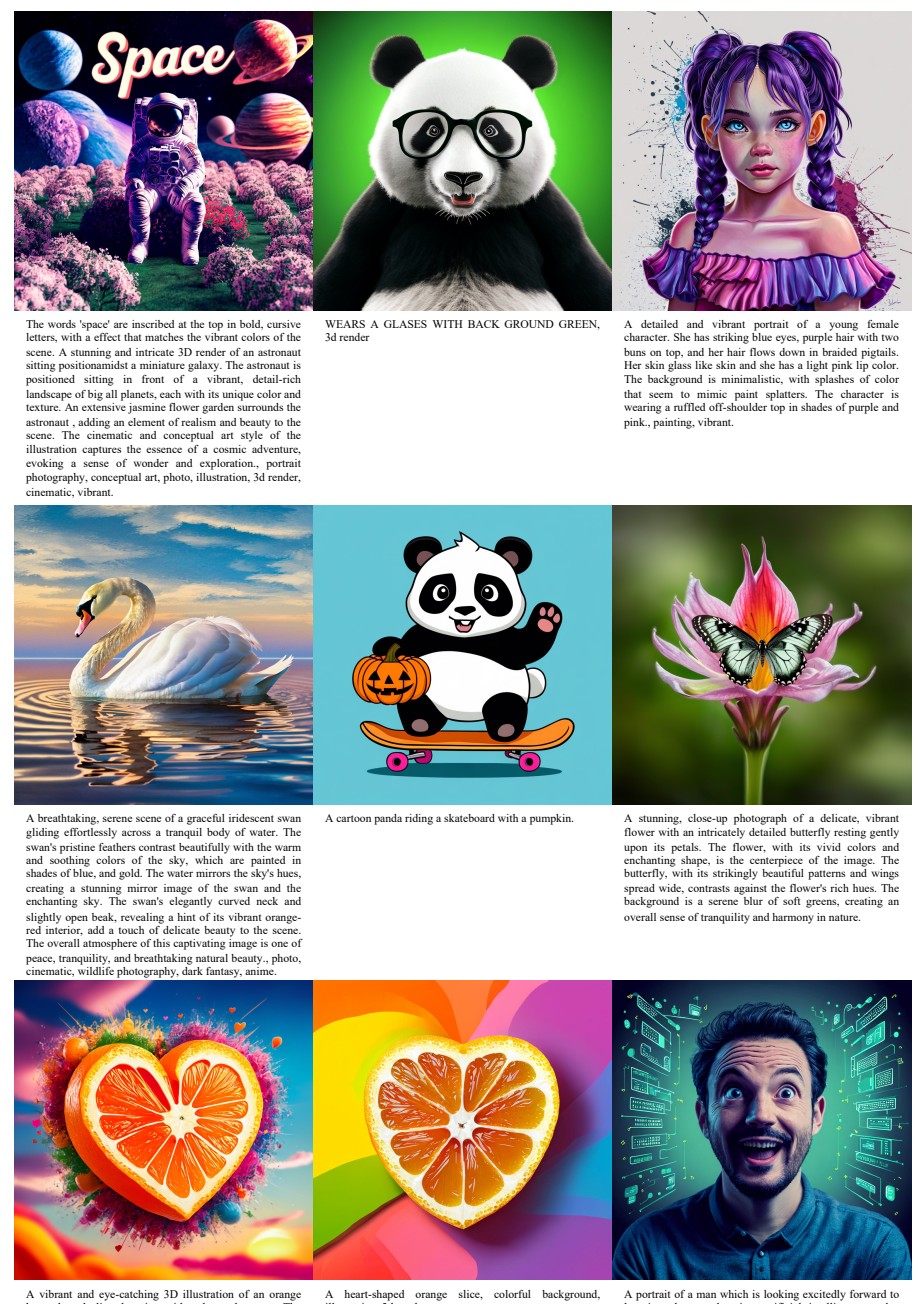

Figure C: Additional results 1/3. Images generated by the Edgen.

the upper bound to 300K to ensure a more sufficient learning of much powerful generation abilities of the most advanced models. The initial text prompts are randomly selected from both the SAM captioning dataset and the community-sourced text prompts.

# G   Additional Results

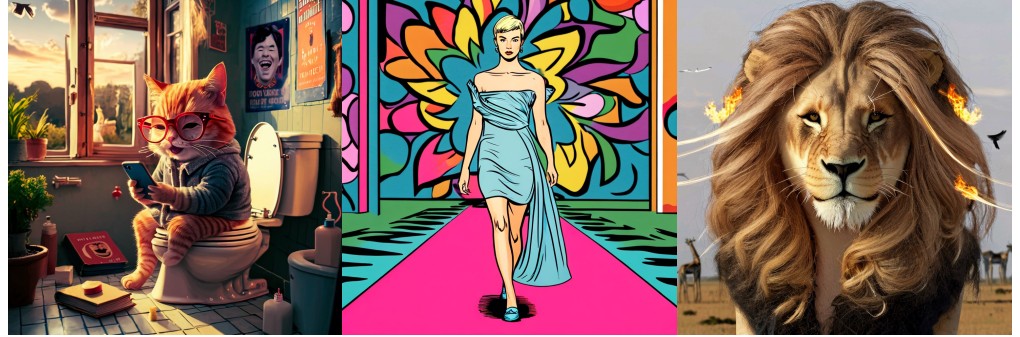

A strange high-definition photo of a cute orange cat, wearing red-rimmed glasses and engrossed in his smartphone, sitting on the toilet. The bathroom was decorated with funny posters, including one of a man laughing and another of a woman. The golden hour sun casts a warm, inviting light through the open windows, illuminating the view. Various items such as books, toilet brushes and soap scattered around add to the unique atmosphere. Potted plants on the floor and natural light streaming in bring a touch of freshness to this cheerful 8k image.

A colorful comic-style linocut illustration of a captivating pop art-inspired composition, a female model walking on a pink carpeted runway. She is wearing a unique, off-shoulder dress with a draped design that exposes parts of her body. The dress is in a light blue color and has a sheer, transparent section at the bottom. The model has a short, blonde hairstyle and is looking directly at the camera. The background is a mesmerizing blend of multi-colored abstract shapes and mandala patterns, creating a harmonious and visually striking scene.

The head of a lion with majesty and elegance. Chin high and pride. Her hair is long and voluminous, and at its ends you can see trails of light and fire coming out of her hair. In the background you can see the Savannah with a few animals in the distance such as giraffes or birds.

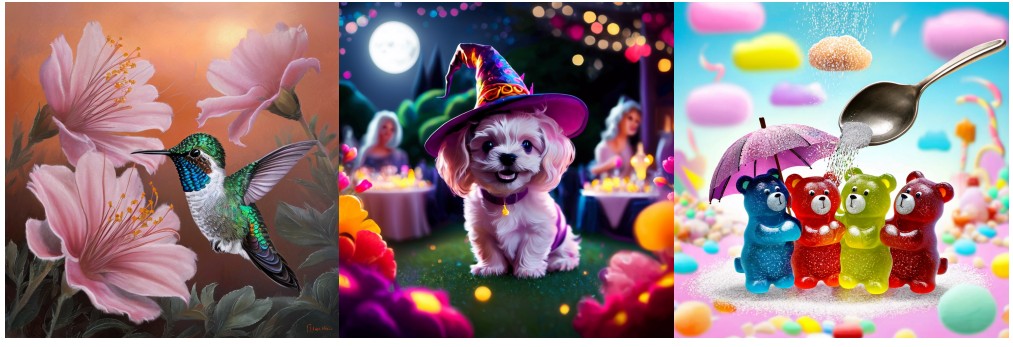

Illustrate a delicate scene where a small, intricately detailed hummingbird with a speckled blue-green throat hovers near a cluster of large, soft pink flowers. The setting is dreamlike, with a misty, orange-toned background suggesting either dawn or dusk. Focus on the light reflecting off the bird's iridescent feathers and the gentle sway of the flower petals., painting.

A charming and captivating illustration of a Maltese puppy wearing a colorful, enchanting witch's hat, attend a glamorous garden party under the moonlight. The puppy is smiling and enjoying the warm atmosphere, surrounded by vibrant, glowing flowers. The background is filled with colorful bokeh, and the scene is bathed in a soft glow. Sharp details and a shallow depth of field create a sense of depth and dimension. The overall atmosphere is cozy and cinematic, with a touch of fairy tale enchantment.

An amusing and whimsical image of four multicolored gummy bears gathered under a small umbrella. A large spoon hovers above them, pouring a shower of sugar over the gummy bears, creating a sweet and playful scene. The background is a bright and colorful dreamscape, filled with an assortment of candies, pastel clouds, and delightful surprises.

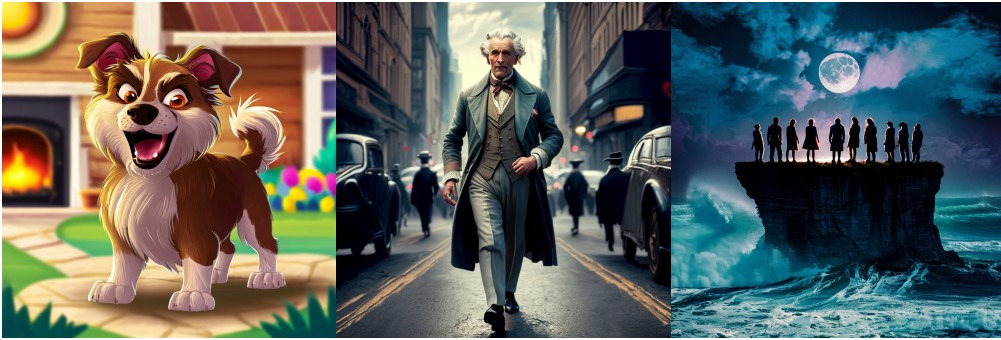

A charming illustration of a dog with its mouth open, showing off its playful and friendly personality. The dog has thick fur and a mischievous glint in its eyes, with a wagging tail that exudes joy. The background features a cozy home with a warm fireplace, a sunny yard, and a colorful garden., illustration.

A breathtaking photo-realistic image of a confident and alluring old classy gentleman, striding down a bustling city street. Dressed in an elegant XIX century suit."

A captivating, cinematic photorealistic image of a group of people standing on a dramatic cliff, their silhouettes accentuated by the soft glow of the full moon. The ocean below is a raging sea with towering waves, creating a sense of awe and danger. The sky is a rich blend of deep blues and purples, with the moon casting a shimmering reflection on the water. The overall atmosphere is eerie yet mysterious, as if the group is witnessing a rare lunar event., photo, cinematic.

Figure D: Additional results 2/3. Images generated by the Edgen.

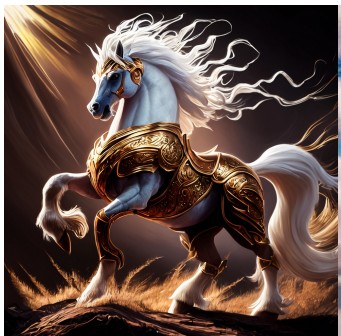 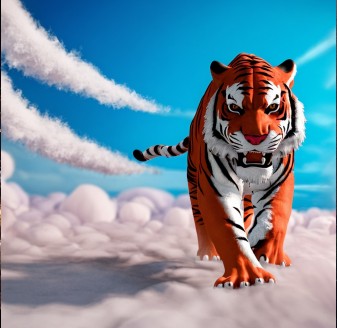 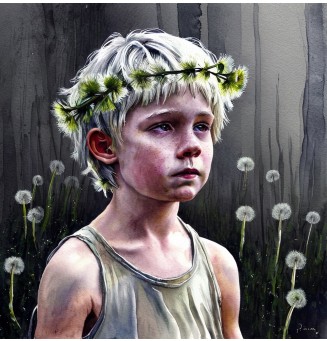

Majestic white-maned stallion, powerful stance, intricate golden plated armor, mythical aura, flowing mane and tail, regal posture, detailed armor engravings, elegant and fierce, dynamic pose, dramatic lighting casting shadows, vibrant contrast between brown and gold, fantasy setting, epic fantasy illustration, high-resolution digital art, Artstation showcase.

A striking and dynamic 3D render of Mount Pleasant High School's tiger mascot, walking determinedly towards the viewer on the right side of the screen. The tiger, has a focused, fierce expression as it strides forward, giving off an air of strength and resilience. The tiger is realistic. The background showcases a vibrant blue sky with soft, cotton-like clouds gently drifting by, creating a serene and motivational atmosphere. The tiger appears to be walking on clouds, giving the image a surreal and uplifting feel., photo, 3d render.

Close-up portrait, the watercolor loose dreamy young boy with platinum blonde hair, adorned with a wreath of dandelions. His serene, contemplative expression conveys a sense of introspection and melancholy. The boys simple sleeveless clothe contrasts with the somber background, creating a striking visual effect., lets nature take its course, soft, light coloring, side view. Damien Hirst, creative Commons attribution, painted in a dainty neutral style with soft light colors and few details.Art by Quentin Blake, Alberto Vargas, Zdzislaw Beksinski, painting, dark fantasy, illustration, anime, conceptual art, architecture.

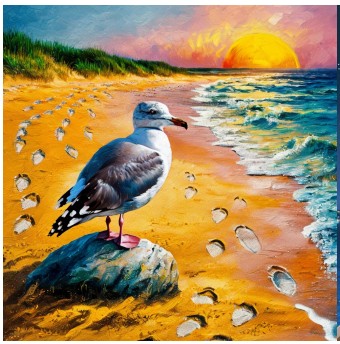 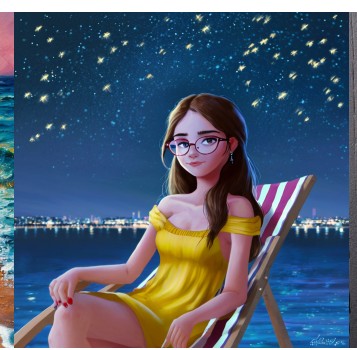 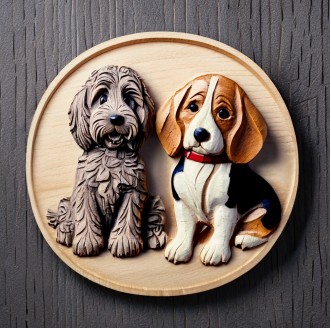

A stunning beach art print capturing a serene coastal scene. In the foreground, a meticulously painted seagull is perched on a rock, its wings outstretched and eyes fixed on the horizon. The golden sand stretches out, dotted with footprints that seem to disappear into the distance. A vibrant sun is setting on the right, casting a warm, golden glow on the waves that gently lap at the shore., painting.

A serene and romantic image of a young woman with long brown hair and glasses, seated on a striped beach chair. She wears a yellow off-shoulder dress that complements her calm demeanor. The starry night sky above her is dotted with countless twinkling stars, and a distant cityscape adds a soft glow to the scene. The reflection of city lights on the water creates a tranquil and intimate atmosphere, perfect for contemplation.

A meticulously carved wooden relief depicting a little cute greydoodle puppie and a cute beagle.

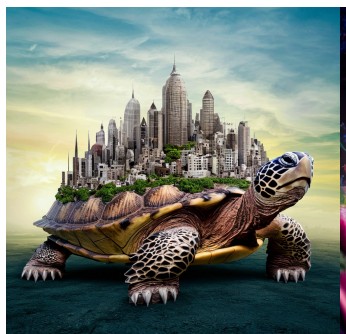 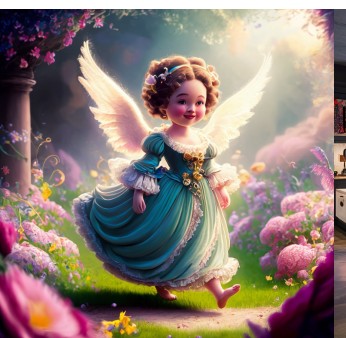 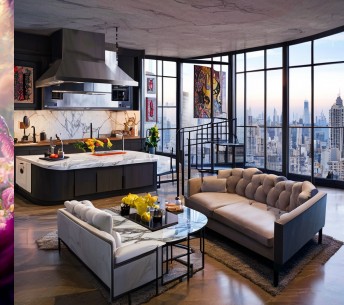

A cityscape built on the back of a giant turtle.

2.5D CG, Charming, Rococo angel: Imagine a children's book illustration of a sweet angel in a flowing Rococo dress. Delicate wings flutter softly amidst a vibrant garden of blooming flowers. Her warm smile radiates as she strolls through a magical heaven, straight out of a fairytale. seed# 33663366, illustration, conceptual art.

In a chic loft apartment, a modern kitchen with stainless steel appliances and a marble island, an open living space with a plush sofa and eclectic artworks, floor-to-ceiling windows overlooking the city, and a spiral staircase leading to the bedroom.

Figure E: Additional results 3/3. Images generated by the Edgen.

