# OpenReview forum: "EvolveDirector: Approaching Advanced Text-to-Image Generation with Large Vision-Language Models"
_NeurIPS.cc/2024/Conference — NeurIPS 2024 poster_

### Official Review · Reviewer_9FQn · 2024-07-11

**Soundness:** 3
**Presentation:** 3
**Contribution:** 3
**Rating:** 5
**Confidence:** 4

**Summary:**

The paper introduces "EvolveDirector," a novel framework designed to train a text-to-image generation model that can compete with advanced models using only publicly available resources. The authors aim to address the limitations posed by proprietary data and the inaccessibility of parameters in state-of-the-art models by leveraging public APIs to obtain text-image pairs for training.

The core of EvolveDirector relies on the interaction with advanced models through their APIs to generate data for training a base model. A significant challenge is the need for a large volume of data (10 million samples or more), which is both time-consuming and computationally expensive. To mitigate this, the authors employ pre-trained large vision-language models (VLMs) to dynamically update and refine the training dataset through various operations like discrimination, expansion, deletion, and mutation.

The framework's efficiency is demonstrated through experiments that show a significant reduction in the required data volume, with only 100k samples needed for comparable performance to models trained on 11 million samples. Furthermore, the paper claims that their final trained model, "Edgen," outperforms several advanced models and will be open-sourced for downstream tasks.

The paper is well-written and presents a strong case for the capabilities of EvolveDirector. It would be beneficial to see additional experiments or comparisons with other state-of-the-art models to further establish the framework's superiority. Moreover, the paper could delve deeper into the specific mechanisms of knowledge transfer from the VLMs to the base model.

**Strengths:**

The paper's core strength lies in its innovative approach to overcome the limitations posed by proprietary data and models. By leveraging publicly available APIs and pre-trained vision-language models (VLMs), EvolveDirector offers a feasible solution to access advanced text-to-image generation capabilities in an open-source environment.
The trained model, Edgen, shows impressive performance, outperforming several advanced models in various aspects. This speaks to The paper includes extensive experiments and ablation studies that validate the framework's effectiveness. The use of human evaluation for model comparison adds a layer of qualitative analysis to the quantitative results.
The authors acknowledge potential biases in the generated content and the importance of safety. They propose integrating debiasing methods and human feedback, showing a responsible approach to AI development.
 The methodology is technically sound, with a clear explanation of the framework's components and the role of VLMs in guiding the evolution of the base model. The paper also details the training strategies and hyperparameter settings, contributing to its reproducibility.
The paper discusses the broader impacts of the work, including potential positive societal impacts such as revolutionizing digital media creation and negative impacts like the risk of bias and misinformation. This demonstrates a well-rounded consideration of the technology's societal effects.

**Weaknesses:**

1. The layout needs to be adjusted. For example, Figure 1 takes up an entire page.
2. The research question 1 & 2 is insightful, which is an important research topic in the perspective of distillation and data annotation.. However I see limited analysis or exploration for this question. Only one result from the trained DiT model to PixART may not be enough as the answer to question 1. As the authors motivate this work with question 1 & 2, introducing more related analysis would be helpful.
3. It seems like the data construction pipeline could be improved by introducing images in reality as [1]. Given that you use VLMs for data refinement, it may be beneficial to use images in reality as ground truth and use VLMs to generation prompts.
4. In Section 4.3, models trained without Discrimination & Expansion & Mutation (first line) can gain comparable results as models trained with Discrimination & Expansion & Mutation (last line). This may cause confusion, as it may lead to a questions: would it be necessary to perform Discrimination & Expansion & Mutation? Would it be all we need to just query advanced generative models to get data?

[1] Zhao, Haozhe, et al. "UltraEdit: Instruction-based Fine-Grained Image Editing at Scale." arXiv preprint arXiv:2407.05282 (2024).

**Questions:**

See weakness.

**Limitations:**

Yes, this paper has limitation section.

---

> ### Author Rebuttal · Authors · 2024-08-07
>
> Thank you for acknowledging that our approach is innovative, "EvolveDirector offers a feasible solution to access advanced text-to-image generation capabilities in an open-source environment", "The trained model, Edgen, shows impressive performance", "The paper includes extensive experiments and ablation studies ",  and "The methodology is technically sound, with a clear explanation ...", etc.
>
> We will address your concerns as follows.
>
> ----
>
> **W.1**:
> In the revision, we will consider shrinking the space occupied by Figure 1, and use the saved space to add more discussions and analyses.
>
> **W.2**:
> Thank you for acknowledging the two research problems. Generating large-scale data based on the naive pipelines to train the base model is extremely costly and time-consuming (hundreds of A100 GPU days for training the base model on million-scale data), let alone training multiple base models to approach multiple advanced models. To tackle this question more efficiently, we simplify the task by fixing the same base model and advanced model to examine the impact of different data scales on the performance of training the base model to approach the advanced model, as shown in the 7 sets of experimental results in Table 2. The results show that the generative capabilities of the advanced model can be learned through training on its large-scale generated data, which is over 11 million. However, if the training data is reduced to 10% or even 1% of the original amount, the performance of the trained model will significantly decrease. The proposed EvolveDirector can train the model in a data-efficient manner, thus offsetting the decrease in model performance caused by reducing the amount of training data.
>
> Regarding question 2, the base model can gain better performance by learning to approach multiple advanced models simultaneously, thus providing an answer to this question. This success is attributed to VLM selecting the best ones from generated images of multiple advanced models as training data.
>
> We will add the discussions and clarification in the revision.
>
> **W.3**:
> Thanks for your comment. Introducing real images as base images and modifying them to generate more training samples is an interesting idea. The mentioned work [1] is suitable to benefit the instruction-based image editing task. However, when pre-train a T2I foundation model, this pipeline may struggle to generate imaginative samples since it creates new samples based on editing the real ones. Besides, in the paper, we focus on exploring how to approach the advanced models based on their generated data. The experimental results demonstrate that relying solely on generated data is sufficient to approach advanced models.
>
> We will add an introduction to this reference [1] in the related works.
>
> **W.4**:
> Please note that the data volumes of these two training methods are significantly different, as shown in Table 2. With Discrimination & Expansion & Mutation, the base model can achieve similar generation performance on a much smaller data scale. We will highlight this to avoid the confusion.
>
> ----
> [1] Zhao, Haozhe, et al. "UltraEdit: Instruction-based Fine-Grained Image Editing at Scale." arXiv preprint arXiv:2407.05282 (2024).

---

### Official Review · Reviewer_i5a7 · 2024-07-12

**Soundness:** 4
**Presentation:** 4
**Contribution:** 4
**Rating:** 8
**Confidence:** 5

**Summary:**

This paper investigates how to train the text-to-image model comparable to advanced models using publicly available resources. Specifically, EvolveDirector collects training data with the APIs of advanced models, and further uses a VLM to continuously refine the training dataset. The proposed VLM refinement significantly reduces the data volume needed to teach a base T2I model and improves the training efficiency.

**Strengths:**

1. The proposed framework is novel, and can significantly reduce the volume of data required to approach the SOTA text-to-image generation performance.

2. This paper investigates one interesting direction: training the base T2I model to approach the advanced models using their generated data, which can be crucial for bridging the gap between public T2I models and close-sourced T2I models.

3. The way of utilizing VLMs is interesting and effective. The VLM dynamically maintains the training dataset to achieve efficient training.

4. Edgen shows good human evaluation performance on generation faithfulness to text prompts, especially in multi-object generation and text generation.

5. The paper is well-organized and technically clear.

**Weaknesses:**

1. The explanation of how to instruct VLM is not detailed. For example, how to ensure the generated text prompts can be parsed correctly and free of errors, which could introduce noise into the training data.

2. It would be better to also evaluate the final model on other benchmarks (e.g., DSG, TIFA).

3. Missing references:
(1) DreamSync: Aligning Text-to-Image Generation with Image Understanding Feedback
(2) SELMA: Learning and Merging Skill-Specific Text-to-Image Experts with Auto-Generated Data

**Questions:**

Typos: in line 336, “outperform” -> “outperforms”.

---

> ### Author Rebuttal · Authors · 2024-08-07
>
> We appreciate your acknowledgment and comments. We will address your concerns as follows.
>
> Notations: W: Weakness, Q: Question
>
> -----
>
> **W.1**:
> The detailed instructions for VLM are provided in the supplementary. We structure the outputs of VLM to ensure the generated text prompts can be parsed correctly and free of errors. The diversity of output formats from VLM can pose challenges for automated parsing. We found that by providing specific instructions to the VLM, its output format can be standardized. Specifically, when prompting the VLM to generate more text prompts, we offer instructions such as “Arrange them in the format of the list ["Text description 1", "Text description 2", ...].” This approach directs the VLM to generate outputs in a consistent format.
>
> **W.2**:
> Thanks for the suggestion. TIFA [1] and DSG [2] are automatic evaluation methods designed to call LLMs to generate several questions and utilize the VQA model to answer them, so as to evaluate the alignment of the generated samples with the text prompts. Since DSG is the newest benchmark and more comprehensive. We evaluate the models on DSG and the results are reported as follows. The open-sourced mPLUG-large [3] is selected as the VQA model.
>
> |                             |          |                        |        |                        |       |                        |       |
> |:--------:|:--------:|:--------:|:--------:|:--------:|:--------:|:--------:|:--------:|
> |                             | Base Model | Pixart-$\alpha$ | DeepFloyd IF | Playground 2.5 | Ideogram | Stable Diffusion 3 |  Edgen (ours) |
> | DSG (↑)       | 68.61       | 72.60                | 74.49              | 74.98                   |79.97                | 80.05             |              80.61              |
>
> **W.3**:
> Thank you for providing these references. We will discuss these works in the related works section to strengthen it. These two works both aim to motivate T2I models to learn from their self-generated images. DreamSync [4] is proposed to improve the T2I models by selecting their own generations and fine-tuning them on the selections. SELMA [5] utilizes the LoRAs to fine-tune the T2I model on different skill-specific image-text pairs, and then merges these LoRAs to build a unified model.
>
> **Q.1**:
> Thanks for pointing this out. We will do our best to check and correct the typos in the paper.
>
> ----
> [1] Hu, Yushi, et al. "Tifa: Accurate and interpretable text-to-image faithfulness evaluation with question answering." Proceedings of the IEEE/CVF International Conference on Computer Vision. 2023.
>
> [2] Cho, Jaemin, et al. "Davidsonian scene graph: Improving reliability in fine-grained evaluation for text-image generation." arXiv preprint arXiv:2310.18235 (2023).
>
> [3] Ye, Qinghao, et al. "mplug-owl: Modularization empowers large language models with multimodality." arXiv preprint arXiv:2304.14178 (2023).
>
> [4] Sun, Jiao, et al. "Dreamsync: Aligning text-to-image generation with image understanding feedback." Synthetic Data for Computer Vision Workshop@ CVPR 2024.
>
> [5] Li, Jialu, et al. "SELMA: Learning and Merging Skill-Specific Text-to-Image Experts with Auto-Generated Data." arXiv preprint arXiv:2403.06952 (2024).

---

> > ### Comment · Reviewer_i5a7 · 2024-08-12
> > **Final Rating**
> >
> > Thanks for the response. The additional experiments address my concerns. I'll keep my rating as strong accept (8).

---

> ### Author Response · Authors · 2024-08-14
> **Reply to Reviewer i5a7**
>
> Dear Reviewer i5a7,
>
> Thank you for your recognition and for providing insightful comments.
>
> Best regards,
>
> Authors of Paper 1029.

---

### Official Review · Reviewer_gzrb · 2024-07-13

**Soundness:** 3
**Presentation:** 2
**Contribution:** 3
**Rating:** 6
**Confidence:** 4

**Summary:**

This paper explores the effectiveness of training a text-to-image (T2I) model using synthetic examples generated by existing T2I models. The authors find that on the order of 10M image-text pairs are necessary to approach the quality of a good model like PixArt-Alpha, while using only 1M or 100k examples results in serious deterioration.

To improve sample efficiency, the authors introduce their online learning framework, EvolveDirector, which leverages a vision-language model (VLM) to curate a better online training set. The VLM is used to compare the current trained checkpoint's generated images against the reference T2I model's images, in order to determine which training examples to drop from the training set, and which training example prompts to generate more variants of for further training. Occasionally, entirely new training prompts are introduced to increase diversity. Overall, this results in a training process which matches the quality of PixArt-Alpha according to human comparisons using only 100k examples.

The authors perform further ablations to show the necessity of each part of the framework, and train a model using four different advanced T2I models as targets, resulting in a final T2I model, named Edgen, which surpasses each of the target models in overall human preferences.

**Strengths:**

**Originality.** The paper demonstrates the possibility of a training framework using synthetic data which leverages expert T2I models and VLMs to maximize data efficiency. This opens up an avenue of research into better use of expert models for informing the training procedure in T2I.

**Quality.** In certain areas, the authors perform thorough ablations to determine the best procedures, and to provide concrete numbers to compare against. For example, they evaluate the candidates for an expert VLM along several axes to justify their choice, and train baseline T2I models at several data scales to demonstrate the performance drop with decreasing scale.

**Clarity.** The high-level approach and framework are explained clearly.

**Significance.** The paper shows that existing T2I model performance can be replicated using only about 1% of the data needed with naive knowledge distillation.

**Weaknesses:**

**Quality.** While most of the experimental details are explained thoroughly, the paper omits any description of the text data used. The text data is a crucial part of the proposed framework, since it is used to seed the training data, as well as for the mutation operation, presumably. There is also no mention of the source of the data used for evaluation. Along the lines of evaluation, the dataset used to compute FID score should also be mentioned.

**Clarity.** The paper fails to clarify many useful details earlier in the paper (abstract + introduction), leaving the overall understanding of the method vague until reaching Section 3. For example, the key components of expansion, deletion, and mutation are mentioned but not explained at all in the introduction.

**Questions:**

1. What is the source of your training data? The training images are generated by the expert/advanced T2I models, but where do the input prompts come from?
2. Following Q1, how do you know the seed prompts for training (i.e. the initial training set) cover a diversity of prompts? While using LLMs to create variations of prompts can result in new combinations of objects or concepts, the resulting new prompts are unlikely to capture the same diversity of text present in natural datasets.
3. Where does the *mutation* operation source the new prompts from? Are they also generated by the same VLM, or taken from the same source as your initial training data? If they come from an existing dataset, what is the benefit of adding them midway through the training process instead of just starting with a initial training set?
4. To my understanding, LLaVA-NeXT (and many other current VLMs) are not trained on examples involving multiple images. Are the images for comparison concatenated or passed separately to LLaVA? Do you notice any statistical confounders, such as choosing the first image more often than the second?
5. The instructions for "expansion" to the VLM, to generate new variations on the prompt, specify only to replace nouns in the prompt. Why is this the strategy used to generate similar prompts? What happens if the T2I model is struggling to generate a specific noun, and the syntactic structure of the prompt is not the weak point? In that case, it feels like the new prompts would not be relevant for further training.
6. When selecting which VLM to use, you say "The output [of the VLM] is scored by 0 for wrong or 1 for correct by human raters". Why not ask the human raters to rank the generations themselves, and then compute how many examples the VLM and the humans agreed on? Having the human raters rate the VLM output seems like it could induce bias in the raters.

**Limitations:**

The authors note that their method could result in a T2I model which inherits biases from both the teacher T2I model and the VLM used for evaluation. They also describe the potential positive and negative broader societal impacts of the research, including improving model training costs and potential for misuse.

---

> ### Author Rebuttal · Authors · 2024-08-07
>
> Thank you for your recognition and for providing detailed comments. We will address your concerns as follows.
>
> Notations: W: Weakness, Q: Question
>
> -----
>
> **W.1 & Q.1**:
> We will incorporate more details about the source text prompts in the revision. Initially, the text prompts are randomly selected from both the SAM captioning dataset, aiming to capture the diversity of natural data, and the community-sourced text prompts, to ensure alignment with user preferences and interests. The ratio of samples from these two resources is equal to each other. These text prompts will be made available for public access. For evaluation and FID calculations, the text prompts are also sampled from these two sources and exclude the ones sampled for training. Below are some randomly selected examples of the text prompts.
>
> SAM Captions:
>
> - "A woman and a young child, likely a toddler, standing together in a dimly lit area. The woman is wearing a headscarf, and the child is wearing a striped shirt. The woman is holding the child close to her, possibly providing comfort or protection. The overall atmosphere of the image is warm and intimate, showcasing the bond between the woman and the child."
>
> Community-sourced text prompts:
>
> - "Text, "Deisy Yaneth" in one line, one word, no spaces, creative colorful sign made of watter and smoke, black background, artistic,ultra realistic, splash effect, incredible details, rich colors, high quality, in the style of Dan Mumford and Daniel Merriam, 3D rendering, poster, photo, fashion, cinematic, typography, 3d render"
>
> - "A realistic cat, act as a nurse, give a injection to me with syringe"
>
> **W.2**:
> Thanks for this comment. We will add more details and explanations in these sections.
>
> -----
>
> **Q.2**:
> As mentioned in the response to W1&Q1, at the beginning of training, we sample the text prompts both from the SAM captions and community-sourced text prompts, to ensure a broad coverage of prompt diversity. Based on this, the VLMs are encouraged to generate more diverse text prompts, such as the example shown in Fig. 3. Besides, as shown in Table 1, the human raters agree with the high diversity of the generated text prompts. It is worth highlighting that there is no strict need to capture the distribution of text prompts in existing datasets, such as the SAM and LAION, where the long tail distribution, etc., may not be the most beneficial for training. Our proposed method aims to adaptively generate the samples that are high-value for training.
>
> **Q.3**:
> We agree with you that if new prompts come from an existing dataset, there is no extra benefit. When developing our method, we took this into consideration and designed the mutation operation to create completely new text prompts using the VLM, without relying on existing ones. The mutation is operated in the training process to encourage the model to explore and learn from a broader domain of text prompts.
>
> **Q.4**:
> We concatenate the two images to form a single contrast question input for the VLM. In our concatenating and comparing manner, as shown in Table 1,  LLaVA-NeXT achieves a similar performance as the Qwen-VL-Max, which is able to interact with multiple images. We arrange the two images in a random order to avoid any potential bias in selection. In our statistical validation, no location bias was found in the selections of LLaVA-NeXT. The results in the "Discrimination" column in Table 1 also show that the selections of LLaVA-NeXT match human raters closely.
>
> **Q.5**:
> Please note that the expansion is to generate more text prompts and will not remove the original ones. If the T2I model is struggling to generate images aligned with the text prompts, the original text prompts will continuously be involved in the training. Besides, whether the new prompts are relevant for further training is determined by the VLM based on their value for learning.
>
> **Q.6**:
> We agree with you that ranking the generations by human raters is an alternative way for evaluations. But in practice, human raters not only evaluate the discrimination, but also evaluate expansion accuracy and diversity, and the latter two are evaluated by scoring outputs of VLM. To streamline the evaluation process and ensure a consistent execution way for human raters, we choose the evaluation approach outlined in the paper. To eliminate potential bias, each output of VLMs is independently evaluated by five human raters, and the types of VLMs are anonymous to them.

---

> > ### Comment · Reviewer_gzrb · 2024-08-12
> >
> > Thank you for your response. I would appreciate adding more details into the paper about the sources of data for all uses, as you explained in response to Q1. In addition, I believe the paper would benefit from more details on the Mutation operation, since at the moment there is only one line describing the operation. For example, what prompt(s) do you use to get the LLM to generate entirely new T2I prompts?
> >
> > Assuming this information is added to the paper/appendix, I am willing to raise my score to 7.

---

> ### Author Response · Authors · 2024-08-12
> **Reply to Reviewer gzrb**
>
> Dear Reviewer gzrb,
>
> Thanks for your valuable feedback. Yes, we will add the details in the response and provide more explanations of mutation in the main paper. The detailed prompts we use to instruct the VLM will be provided in the appendix. Thanks again for your comments on improving our paper.
>
> Best regards,
>
> Authors of Paper 1029.

---

> ### Author Response · Authors · 2024-08-14
> **Reply to Reviewer gzrb**
>
> Dear Reviewer gzrb,
>
> To further address your concerns, we provide detailed text prompts for mutation as follows, which will be included in the revised paper.
>
> Text Prompt:
>
> 'Now, exercise your imagination to generate 1 new text description for visual contents, {*enhanced_prompt*}. It should be completely unrelated to the previous images and have a completely different structure from the previous text descriptions. Arrange it in the format of a list ["Text description"].'
>
> The *enhanced_prompt* is a prompt controlling the length and the granular level of the generated samples.
>
> The *enhanced_prompt* is randomly sampled from the following ones：
> - 'which should contain less than 10 words and be rough and short'
> - 'which should contain less than 30 words with different granular levels of details'
> - 'which should contain over 30 words with a lot of details'
>
> Three corresponding generated results are as follows：
>
> - ["The sun rises over a calm sea."]
>
> - ["A lone adventurer stands at the edge of a cliff, gazing into the distance, with a single white bird flying overhead and a black raven perched on a nearby rock."]
>
> - ["In the serene setting of a lush garden, a group of vibrant flowers and a variety of exotic fruits coexist harmoniously. The garden is teeming with life, from the delicate petals of the flowers to the succulent fruits hanging from the trees. The colors are a feast for the eyes, with the reds, blues, and yellows of the flowers contrasting beautifully against the green foliage. The fruits, in shades of red, orange, and yellow, add a pop of color to the scene. The garden is a symphony of nature, where every element has its place and purpose."]
>
> Best regards,
>
> Authors of Paper 1029.

---

### Author Rebuttal · Authors · 2024-08-07

We would like to express our gratitude to the reviewers for their insightful comments. We appreciate the recognition of the strengths of our paper, including the novelty and effectiveness of the proposed EvolveDirector, and the recognition of extensive experiments and ablation studies.

We would like to highlight the following recognitions from the reviewers：

- Reviewer gzrb: "This opens up an avenue of research into better use of expert models for informing the training procedure in T2I."

- Reviewer i5a7: "The proposed framework is novel" and "The way of utilizing VLMs is interesting and effective."

- Reviewer 9FQn:
"EvolveDirector offers a feasible solution to access advanced text-to-image generation capabilities in an open-source environment", "The trained model, Edgen, shows impressive performance", and "The paper includes extensive experiments and ablation studies that validate the framework's effectiveness."

We address the concerns of reviewers in detail in the subsequent responses respectively.

---

### Comment · Area_Chair_Y2Xw · 2024-08-12

Dear reviewers: as you are aware, the reviewer-author discussions phase ends on Aug 13. We request you to kindly make use of the remaining time to contribute productively to these discussions. If you have not read and/or responded to author rebuttal, please do it asap so that the authors get a chance to respond to you. If you have more questions to ask or want further clarification from the authors, please feel free to do it.

---

### Decision · Program_Chairs · 2024-09-25

**Decision:**

Accept (poster)

**Comment:**

The work uses vision-language models to augment text-to-image generation. The combination of these two kinds of models is interesting. All reviewers agree that the paper can be accepted. The author responses solved most questions, which can be included in the final version.